# Sensitive Quantitative In Vivo Assay for Evaluating the Effects of Biomolecules on Hair Growth and Coloring Using Direct Microinjections into Mouse Whisker Follicles

**DOI:** 10.3390/biom13071076

**Published:** 2023-07-05

**Authors:** Lipeng Gao, He-Li Zhang, Xiao-Yang Tan, Yan-Ge Wang, Hongzhi Song, Vicky Lan Yuan, Xin-Hua Liao

**Affiliations:** 1School of Life Sciences, Shanghai University, Shanghai 200444, China; 2School of Medicine, Shanghai University, Shanghai 200444, China; 3School of Environmental and Chemical Engineering, Shanghai University, Shanghai 200444, China

**Keywords:** hair loss, abnormal skin pigmentation, minoxidil, ruxolitinib, RSPO1, deoxyarbutin, NBL1, vibrissae follicle, whisker follicle microinjection assay, drug development

## Abstract

Many people suffer from hair loss and abnormal skin pigmentation, highlighting the need for simple assays to support drug discovery research. Current assays have various limitations, such as being in vitro only, not sensitive enough, or unquantifiable. We took advantage of the bilateral symmetry and large size of mouse whisker follicles to develop a novel in vivo assay called “whisker follicle microinjection assay”. In this assay, we plucked mouse whiskers and then injected molecules directly into one side of the whisker follicles using microneedles that were a similar size to the whiskers, and we injected solvent on the other side as a control. Once the whiskers grew out again, we quantitatively measured their length and color intensity to evaluate the effects of the molecules on hair growth and coloring. Several chemicals and proteins were used to test this assay. The chemicals minoxidil and ruxolitinib, as well as the protein RSPO1, promoted hair growth. The effect of the clinical drug minoxidil could be detected at a concentration as low as 0.001%. The chemical deoxyarbutin inhibited melanin production. The protein *Nbl1* was identified as a novel hair-growth inhibitor. In conclusion, we successfully established a sensitive and quantitative in vivo assay to evaluate the effects of chemicals and proteins on hair growth and coloring and identified a novel regulator by using this assay. This whisker follicle microinjection assay will be useful when investigating protein functions and when developing drugs to treat hair loss and abnormal skin pigmentation.

## 1. Introduction

Human hair has numerous functions, including thermoregulation, touch sensation, and physical protection. It can also serve as an indicator of body health conditions [1]. Hair loss or excessive hair growth, as well as abnormal skin or hair pigmentations such as melasma and premature hair greying, can lower patients’ self-esteem and cause psychological distress [2,3]. The most commonly used medications to treat hair loss are minoxidil and finasteride [4,5], whereas treatments for abnormal pigmentation include hydroquinone, retinoids, corticosteroids, and light and laser therapy [6,7,8]. However, the effectiveness of these therapies is limited, and they are often associated with undesirable side effects [9]. Therefore, a simple and reliable assay that can be used to evaluate the functions of molecules on hair growth and coloring needs to be developed to facilitate drug development.

Several methods are currently used to evaluate the effects of molecules on hair growth. One method involves adding molecules to the culture media of dermal papilla cells [10,11], outer root sheath cells [12], or skin organoids [13] to examine their effects on cell growth. However, this in vitro cell assay does not accurately reflect in vivo conditions. Another method involves topically applying molecules onto the back skin of mice or subcutaneously injecting molecules with slow-releasing beads into the back skin of mice for several consecutive days and observing the timing of the hair growth phase shift from telogen (the resting phase) to anagen (the growth phase) or vice versa [14,15,16]. These methods are qualitative rather than quantitative and often require a large number of molecules, which can cause the experiments to be expensive. A third method involves adding molecules to the culture media of whisker follicles isolated from mice [14,17,18] or hair follicles isolated from a human scalp [19,20]. This method can be used to quantitatively evaluate the effects of molecules on hair growth by measuring the length of the hair shafts, and it has been widely used in preclinical research. However, remarkable variation exists between individual follicles (in terms of size, growth phase, etc.) [21], and the interaction between hair follicles and surrounding cells (such as immune cells [22,23], adipocytes [24], and nerves [25]) in vivo cannot be recapitulated by using this assay.

The current methods used to evaluate the effects of molecules on hair and skin pigmentation also have various limitations. One method involves using inhibitory activity against tyrosinase, a rate-limiting enzyme that produces melanin, or against melanin production in melanocytes or melanoma cells as an in vitro assay to screen for lead compounds [26,27,28,29,30]. A second method involves using zebrafish as a whole-animal pigmentation model for the phenotype-based screening of melanogenic inhibitors or stimulators [31]. However, using zebrafish as an in vivo model may not accurately reflect the effects on human skin due to the differences between the two species. The third method involves inducing skin pigmentation by irradiating Yucatan swine or guinea pigs with UV light and then observing changes in their skin color by topically applying or subcutaneously injecting candidate substances [6,32]. This method is relatively expensive and time consuming, and quantifying the effects is difficult.

The miniorgan hair follicle is a skin appendage located deep in the dermis. A hair follicle consists of different compartments, including the sebaceous gland, bulge, inner root sheath, outer root sheath, hair bulb, and dermal papilla. In hair bulbs, the hair matrix epithelial cells proliferate to form the hair shaft, whereas the melanocytes transport the melanosomes to the neighboring epithelial cells to deposit the pigments [33]. The dermal papilla regulates hair growth and coloring by releasing secretory factors [11,33,34].

The whisker, also called the vibrissae, tactile hair, or sinus hair, is characterized by a follicle surrounded by two venous sinuses and a rigid fibrous capsule [35,36]. Whisker follicles exhibit an anatomy and molecular signaling that is similar to regular hair follicles, which explains why whisker follicles isolated from mice have been employed in cultures to evaluate the effects of molecules on hair growth in preclinical research [14,17,18].

The whisker follicles are much larger than regular hairs and are usually left–right symmetrical. Taking advantage of these unique characteristics, we developed a new assay called “whisker follicle microinjection assay” to evaluate the effects of molecules on hair growth and pigmentation. With this method, we first plucked the whiskers and then directly injected the test molecules into the whisker follicles on one side by using a microsyringe with a needle that was a similar size to the whisker (~110 μm), which allowed the whiskers on the other side to be used as a control. The length and blackness of the regrown whiskers on both sides were then quantitatively measured and compared in pairs. We successfully demonstrated the advantages of this new method by using known chemicals and proteins. This new method will be useful to investigate protein functions in hair growth and coloring and to develop related drugs.

## 2. Materials and Methods

### 2.1. Animals and Reagents

C57BL/6, ICR mice, and SD rats were purchased from Shanghai Jieshige Laboratory Animal Co., Ltd. (Shanghai, China) and housed under specific pathogen-free (SPF) conditions; they had free access to food and water and were under a 12 h light and dark cycle. All experiments involving animals were performed in accordance with the guidelines of the Institutional Animal Care and Use Committee (IACUC) and were approved by the Ethics Committee of Shanghai University.

Coomassie brilliant blue R250 (0.025 g, A610037, Sangon Biotech, Shanghai, China) was dissolved in 4.5 mL of methanol and 1 mL of glacial acetic acid and added to 4.5 mL of H_2_O. Minoxidil (50 mg, M831481, Macklin, Shanghai, China) was dissolved in 1 mL of glycerol and 1 mL of 75% ethanol and then diluted to the appropriate concentration with 0.9% saline. Ruxolitinib (R849099, Macklin, Shanghai, China) was dissolved in DMSO and then diluted with 20% cyclodextrin. D-arbutin (ID0490, Solarbio, Beijing, China) was dissolved in DMSO and then diluted with 0.9% saline. RSPO1 (50316-M08S) and NBL1 (50976-M08H) proteins were purchased from Sino Biological (Beijing, China). The HaCaT cell line was introduced from the Cell Bank of the Chinese Academy of Sciences (SCSP-5091).

### 2.2. Whisker Follicle Microinjection

Adult male C57BL/6 mice were anesthetized with an intraperitoneal injection of 4% chloral hydrate (200 μL/20 g). The test molecules were then directly injected into the whisker follicles using a microsyringe (NF36BL-2, World Precision Instruments, Inc., Sarasota, FL, USA) with a needle that was a similar size to the whisker (outside diameter = 110 μm) under a dissection microscope (SMZ745T, Nikon, Tokyo, Japan). Completely dissolving the injected molecules was important to avoid clogging the needle. After the injection, we paused for a few seconds before slowly withdrawing the needle to avoid liquid leakage.

### 2.3. Quantitative Analysis of Whisker Length and Blackness

The whiskers were plucked and taped onto white paper. They were straightened as much as possible, and their lengths were measured using vernier calipers. If two whiskers grew from the same follicle, only the longer whisker was counted. The lengths of the paired control whiskers were normalized to one.

The plucked whiskers taped on the white paper were photographed with a stereo microscope and were transformed into greyscale images. Two millimeters of the lightest color in the middle of the whiskers were selected for grey value quantification using Image J software. The grey value of the paired control whisker was normalized to one.

Statistical differences were evaluated using a two-tailed paired Student’s *t*-test (GraphPad Prism v8.0, GraphPad Software, San Diego, CA, USA) and a Wilcoxon signed-rank test (R version 4.1.2). The difference was considered significant when *p* < 0.05 (*), *p* < 0.01 (**), and *p* < 0.001 (***).

### 2.4. RNA-Seq

The total RNA was extracted from back skin samples of the ICR mice at ages P14, P20, P30, and P55, with duplicate samples at each time point, using TRIzol (15596018; Thermo Fisher Scientific, Waltham, MA, USA) according to the manufacturer’s instructions. The cDNA synthesis, RNA-seq library construction, RNA-seq, and sequencing data analysis were performed by the Novogene Corporation (Nanjing, China). The RNA-seq data analysis procedures included raw read processing, clean read mapping and filtering, and the normalization of gene expression levels.

### 2.5. Cell Growth Curve

The log-phase HaCaT cells were seeded in triplicate in the wells of a 96-well plate at a density of 300 cells per well. After a 6 h incubation period to allow for cell attachment, images of each well were captured, and the cell numbers were automatically counted using the Operetta CLS high-content analysis system (PerkinElmer, Waltham, MA, USA). This process was repeated for four consecutive days, and the resulting cell numbers were plotted over time.

### 2.6. qPCR

The total RNA was extracted from the back skin samples of the C57BL/6 mice using TRIzol (15596018, Thermo Fisher Scientific, Waltham, MA, USA), and 1 µg of the total RNA was reverse transcribed with the RevertAid First Strand cDNA Synthesis Kit (K1621, Thermo Fisher Scientific, Waltham, MA, USA) using oligo (dT) primers according to the manufacturer’s instructions. A qPCR was performed on the CFX Opus 96 Real-Time PCR System (Bio-Rad, Hercules, CA, USA) using the Hieff^®^ qPCR SYBR Green Master Mix (11171ES08, Yeasen Biotechnology, Shanghai, China). The fold changes in the gene expression between the samples were calculated using the 2^−ΔΔCT^ method. The housekeeping gene Mbp was used as an internal control to normalize the CT values between the samples. The qPCR primers used were as follows: Nbl1_F, ACAATGCTTCAGTTACAGCGT; Nbl1_R, CAGGGCACTCCAAGGTCAC; Tbp_F, GGCGGTTTGGCTAGGTTT; Tbp_R, GGGTTATCTTCACACAC-CATGA.

## 3. Results

### 3.1. Development of Whisker Follicle Microinjection Assay Based on the Bilateral Symmetry of Mouse Whiskers

Mouse whiskers and follicles were designated according to their positions, which were numbered from A1 to E5 (Figure 1A,B and reference [37]). To confirm the symmetry of the mouse whiskers, the whiskers were plucked, and the lengths were measured with vernier calipers. The length of each whisker on the left side was normalized to one. A Student’s *t*-test revealed that no significant difference existed between the left and right sides (Figure 1C,D and Appendix A). Additionally, after plucking, the regrown hairs were also left–right symmetric, as indicated by the length comparison of the paired whiskers between the left and right sides (Figure 1E,F).

Taking advantage of the bilateral symmetry and larger size of the mouse whisker follicles, we developed a novel assay called a whisker follicle microinjection assay to evaluate the effects of the molecules on hair growth and coloring. For this assay, we plucked the whiskers and then used a microsyringe with a needle that was a similar size to the whisker (outside diameter = 110 μm) to inject the molecules and control solvent directly into the whisker follicles on the left and right sides, respectively (Figure 2A). The molecules diffused around the injected whisker follicles to affect hair growth and coloring, as demonstrated by the Coomassie brilliant blue microinjection (Figure 2B). By comparing the length and color intensity of the whiskers between the two sides, we were able to quantitatively evaluate the test molecules on hair growth and coloring.

### 3.2. The Microinjection of RSPO1 Protein into Whisker Follicles Effectively Promotes Hair Growth

We used known proteins to test our assay. RSPO1 can induce the transition from the telogen to the anagen hair-growth phase via an intradermal injection of the protein into the back skin of midtelogen mice for 7 consecutive days [38]. Five pairs of large whiskers were selected (B2, C1, C2, D1, and D2) and plucked, and 5 μL of a 0.5 mg/mL RSPO1 protein or a PBS buffer were injected via a microsyringe into the whisker follicles on the left and right sides, respectively. The regrown whiskers were harvested, and the lengths were measured (Figure 3A), whereby the length of each whisker on the control side was normalized to one. The average normalized length of the whiskers treated with RSPO1 was approximately 1.8 times longer than that of the control. The results of the data analysis revealed that this difference was statistically significant (*p* < 0.01, Figure 3B). These results confirm the ability of RSPO1 to stimulate hair growth and demonstrate that our assay can be used to quantitatively evaluate the effect of proteins on hair growth.

### 3.3. The Microinjection of Minoxidil or Ruxolitinib Chemicals into Whisker Follicles Effectively Promotes Hair Growth

We then tested our whisker follicle microinjection assay by using the known small chemicals minoxidil and ruxolitinib. Minoxidil, an original antihypertensive drug, has been used as a primary treatment for androgenetic alopecia and other hair-loss disorders [4,39]. Ruxolitinib, a JAK inhibitor, has potential therapeutic effects when used to treat rheumatoid arthritis, vitiligo, and alopecia areata [40,41]. For our assay, five pairs of large whiskers were plucked from adult male mice, and 5 μL of 0.001%, 0.01%, 0.1%, and 1% minoxidil or a control solvent was injected into the follicles via a microsyringe. The regrown whiskers were plucked and imaged and the lengths were measured (Figure 4A and Appendix A). The results of a statistical analysis revealed that minoxidil had stimulatory effects on hair growth in a dose-dependent manner from a dose of 0.001% to 0.1% (Figure 4B). Although the length of the whiskers treated with 1% minoxidil increased in comparison with the control, the length decreased in comparison with the 0.1% dose, which indicated a toxic effect of minoxidil at a high dose (Figure 4B). Similar results were obtained when using ruxolitinib (Figure 4C,D and Appendix A). We also demonstrated that this assay could be used in rats by using 0.1% minoxidil (Appendix A). In clinical practice, a topical application of a 5% minoxidil solution is used to treat alopecia. Our assay, however, can detect the effects of minoxidil (and ruxolitinib) at concentrations as low as 0.001%, which indicates a high sensitivity.

### 3.4. The Microinjection of the Chemical Deoxyarbutin into the Whisker Follicle Effectively Inhibits Melanin Production

We then examined whether our whisker follicle microinjection assay could be adapted to evaluate the effects of molecules on hair pigmentation. Not only the length but also the color of mouse whiskers is left–right symmetric (Figure 5A). The natural compound deoxyarbutin, a reversible tyrosinase inhibitor, has been widely used in skin lighting [42,43,44]. For our experiment, five pairs of large whiskers from adult male mice were plucked, and 5 μL of 3% deoxyarbutin or a control solvent was injected into the follicles. The regrown whiskers were plucked and imaged. The images were transformed into greyscale, and two millimeters of the lightest color in the middle of each whisker was selected for grey-value quantification. The results of a statistical analysis confirmed that deoxyarbutin effectively inhibited melanin production, which indicates that our assay can also be used to evaluate the effects of molecules on hair coloring.

### 3.5. Identification of NBL1 as a Novel Hair-Growth Inhibitor Using Whisker Follicle Microinjection Assay

To investigate the changes in gene expression during the hair growth cycle, we selected four time points—two during the growth phases (P14 and P30) and two during the resting phases (P20 and P55). At each time point, we harvested skin samples from the backs of two mice and used RNA-seq technology to analyze the expression profiles of the skin. The results of our analysis identified numerous secretory factors associated with the hair cycle (the data will be published elsewhere), including the NBL1. *Nbl1* was found to be expressed at low levels during the growth phases and at high levels during the resting phases (Figure 6A), which was confirmed with a q-PCR (Figure 6B). In vitro, recombinant NBL1 added to culture media inhibited the growth of immortalized human keratinocytes HaCaT in a dose-dependent manner (Figure 6C). Furthermore, an in vivo injection of the recombinant NBL1 protein into the whisker follicles significantly inhibited hair growth (Figure 6D,E), which provides evidence that NBL1 is a novel negative hair-cycle regulator.

## 4. Discussion

A simple, sensitive, and reliable assay to evaluate the effects of biomolecules on hair growth and coloring would strongly aid scholars who are conducting functional studies on genes and developing drugs to treat hair loss or skin and hair pigmentation. Currently, methods such as topically applying biomolecules to the back skin of mice or injecting biomolecules with slow-releasing beads into the back skin of mice are used to observe the transition from the telogen to the anagen phase [11,12]. However, these methods are not quantitative and often require high doses of biomolecules and the repeated administration of drugs for multiple days. Another method involves isolating whisker follicles from mice [13,14,15] or hair follicles from a human scalp [16,17], culturing the organs in media, and then adding test biomolecules to the culture media to observe the effects based on the hair elongation length. However, the isolated hair follicles may vary in size and be in different growth phases, which results in high variations between individual follicles and the need for dozens of hair follicles per group to obtain conclusive results [20,21]. This causes the collection of enough samples to be time-consuming and difficult, regardless of if the samples are collected from mouse whiskers or human hair follicles. In addition, this assay, whereby cultured organs are used, cannot be used to evaluate the effects of biomolecules on the interactions between hair follicles and surrounding cells in vivo (such as immune cells, adipocytes, and nerves).

We established a novel in vivo assay called a whisker follicle microinjection assay, which takes advantage of the bilateral symmetry and larger size of mouse whisker follicles. By injecting test biomolecules and a control solvent directly into the whisker follicles on the left and right sides of the whiskers, respectively, and comparing the lengths of the regrown whiskers in pairs, we quantitatively evaluated the effects of biomolecules on hair growth. This assay is highly sensitive, as it can detect the effect of the clinical drug minoxidil at a concentration as low as 0.001% (5000 times lower than the concentration of 5% used in clinical practice) [45]. The JAK/STAT pathways have been proposed to be involved in the T-cell mediated inflammation of the hair follicle microenvironment and the autoimmune disease alopecia areata (AA) [46]. Ruxolitinib, a JAK inhibitor, induces hair growth in patients with moderate-to-severe AA [47]. Our whisker follicle microinjection assay can be used to detect the effect of ruxolitinib at a concentration as low as 0.001%, which indicates that this in vivo assay can also sensitively detect the effects of biomolecules interfering with the interaction between hair follicles and the immune system. The high sensitivity of the assay makes testing small amounts of biomolecules possible, which can be particularly useful if the test biomolecule is hard to produce or is expensive.

We used this assay to demonstrate that the newly identified hair-cycle-associated secretory protein NBL1 has an inhibitory effect on hair growth. The molecular function of NBL1 on the hair cycle is very interesting and is under further investigation.

This whisker follicle microinjection assay can also be adapted to evaluate the effects of biomolecules on hair coloring by quantifying the grey value of the regrown whiskers treated with biomolecules. However, the affected segments of the whiskers should be selected, which can be difficult if the effects on pigmentation are not remarkable enough. The assay is not as sensitive when evaluating the effect of biomolecules on hair growth, so we were not able to demonstrate a dose-dependent effect of deoxyarbutin on whisker pigmentation. This method may be further optimized in the future.

Our steel microneedles are resilient, allowing for easy operation under a dissection microscope. However, steel microneedles can still bend or become clogged. To avoid this, the test biomolecules should be completely dissolved, and filtering the solution further before the injection is helpful. Additionally, using a slightly larger microneedle may help prevent bending and clogging, but this could cause the insertion of the microneedle into the whisker follicles to be difficult. The material used to create the microneedles and their lengths can be further optimized to increase the usability of this assay.

Many mammals have large and bilaterally symmetrical whiskers, and we showed that this assay can be used on rats. This assay can also be used on other animals such as cats, dogs, rabbits, and foxes. In general, larger follicles are preferred, as they allow for an easier microneedle insertion. Large hair follicles on the backs of animals such as pigs and hedgehogs can also be used for the microinjection of biomolecules. In addition to proteins and small chemicals, large entities such as antibodies, viruses carrying genes to be studied, exosomes, and cells can be injected into the whisker follicles to evaluate their functions.

Overall, the whisker follicle microinjection assay we have established here is highly sensitive, quantitative, easy to operate, and accurately reflects in vivo conditions. This novel assay can be used for investigating the functions of proteins or genes in relation to hair growth and coloring. Moreover, it serves as a valuable tool for evaluating the effects of biomolecules on hair growth and coloring. As a result, it will greatly aid in the development of these biomolecules into innovative drugs for treating diseases like hair loss and abnormal skin pigmentation.

## Figures and Tables

**Figure 1 biomolecules-13-01076-f001:**
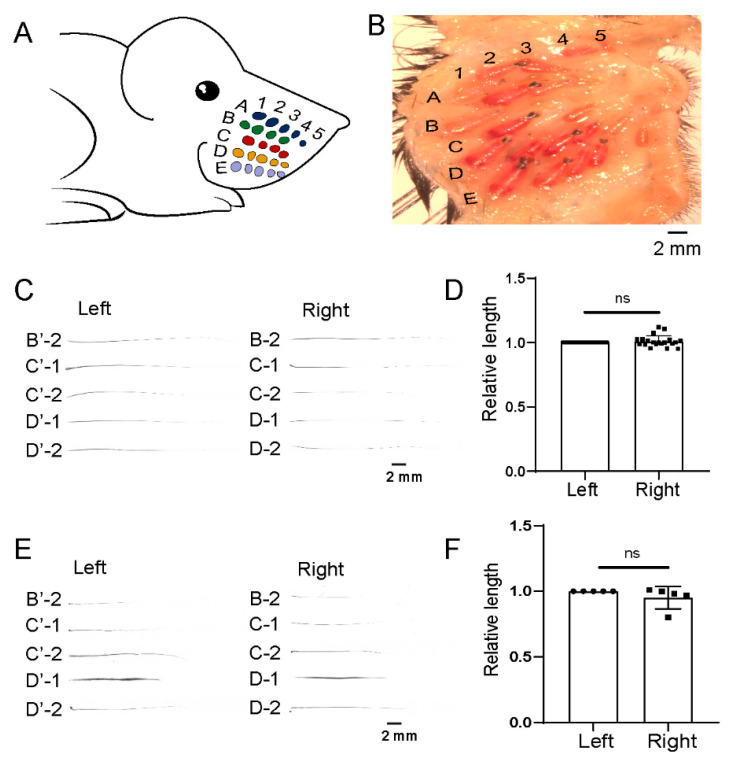
Mouse whiskers are left–right symmetric. (**A**) The diagram shows the layout of the mouse and rat whiskers. Each whisker is represented by a round dot and labeled according to its location (rows A–E and arcs 1–5). For example, the first whisker in the first row is labeled A-1. (**B**) The layout of mouse whisker follicles with dermis side up and the fat, fascia, and underlying muscle tissues removed. (**C**) Five pairs of large whiskers from adult male mice were plucked and imaged. Please see Appendix A for the images of all whiskers. (**D**) The lengths of all whiskers were measured and statistically analyzed (ns, not significant). (**E**) Five pairs of large whiskers from adult male mice were plucked, and the regrown whiskers were plucked again and imaged after 15 days. (**F**) The lengths of five pairs of regrown whiskers were measured, and the statistical differences were evaluated using a two-tailed paired Student’s *t*-test (ns, not significant). Scale bars, 2 mm.

**Figure 2 biomolecules-13-01076-f002:**
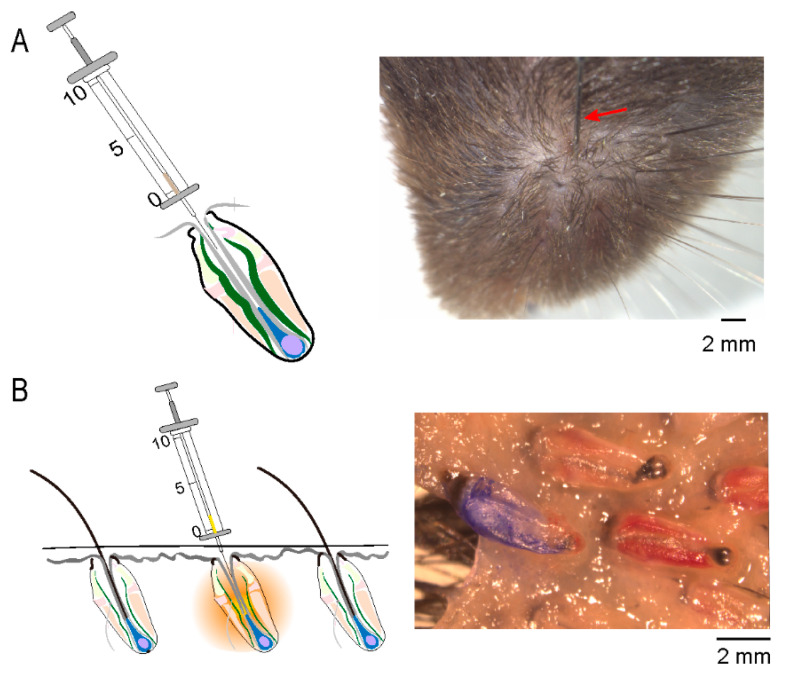
Schematic diagram of whisker follicle microinjection. (**A**) The left panel of the diagram illustrates the structure of a whisker follicle and a syringe with a microneedle inserted into the whisker follicle after hair plucking. The right panel shows a real image of the microneedle (pointed at by the red arrow) being inserted into the whisker follicles. (**B**) The left panel of the diagram illustrates the diffusion of molecules around the whisker follicle injected using a microsyringe. The right panel shows a real image of Coomassie brilliant blue diffused around the injected whisker follicle. Scale bars, 2 mm.

**Figure 3 biomolecules-13-01076-f003:**
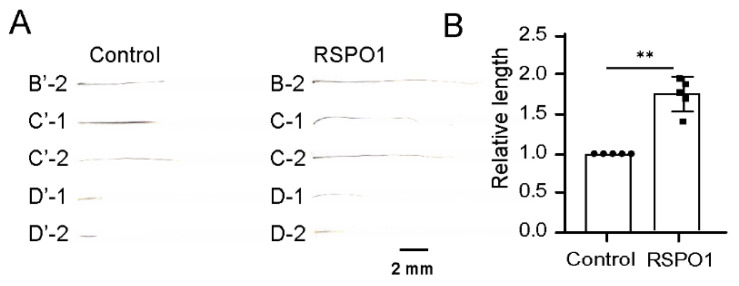
The microinjection of RSPO1 protein into the whisker follicle effectively promotes hair growth. (**A**) Five pairs of large whiskers from adult male mice were plucked, and 5 μL of 0.5 mg/mL RSPO1 protein or PBS buffer were injected via microsyringe into whisker follicles on the left and right sides on days 1, 3, and 5. Each pair of regrown whiskers was harvested and imaged on different days (between days 11 and 20). (**B**) The length of each regrown whisker was measured, and the statistical difference was evaluated using a two-tailed paired Student’s *t*-test. ** *p* < 0.01. Scale bar, 2 mm.

**Figure 4 biomolecules-13-01076-f004:**
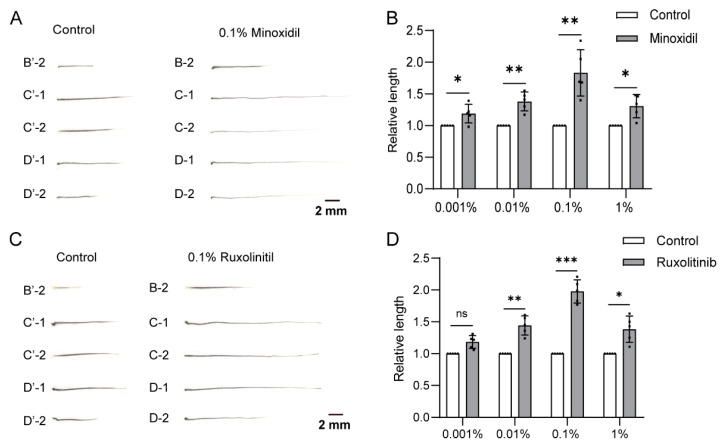
The microinjection of minoxidil and ruxolitinib chemicals into the whisker follicle effectively promotes hair growth. (**A**) Five pairs of large whiskers from adult male mice were plucked, and 5 μL of 0.1% minoxidil or a control solvent was injected into the follicles on the left and right sides via microsyringe on days 1, 3, and 5. On day 12, the regrown whiskers were plucked and imaged. Additional images of microinjections of 0.001%, 0.01%, and 1% minoxidil can be found in Appendix A. (**B**) The lengths of the regrown whiskers treated with different minoxidil doses were measured and statistically analyzed. (**C**) Like in (**A**), the five pairs of regrown whiskers treated with 0.1% ruxolitinib were imaged. Additional images of microinjections of 0.001%, 0.01%, and 1% ruxolitinib can be found in Appendix A. (**D**) The lengths of regrown whiskers treated with different doses of ruxolitinib were measured, and the statistical differences were evaluated using a two-tailed paired Student’s *t*-test at *p* < 0.05 (*), *p* < 0.01 (**), and *p* < 0.001 (***), (ns, not significant). Scale bar, 2 mm.

**Figure 5 biomolecules-13-01076-f005:**
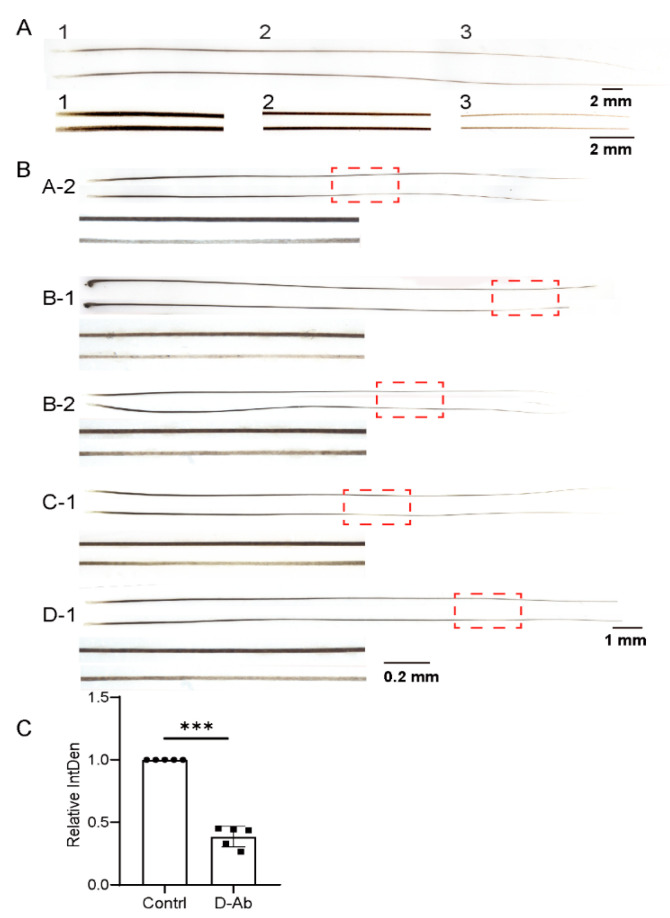
The microinjection of deoxyarbutin chemical into the whisker follicle effectively inhibits melanin production. (**A**) The blackness of mouse whiskers is left–right symmetric. A pair of whiskers without any treatment were imaged. The images of paired whisker shafts at different segments were magnified and compared for color intensity. (**B**) Five pairs of large whiskers from adult male mice were plucked, and 5 μL of 3% deoxyarbutin or a control solvent was injected into the follicles on the left and right sides via microsyringe on days 1, 3, and 5. On day 20, the regrown whiskers were plucked and imaged. The images were transformed into greyscale, and two millimeters of the lightest color in the middle of each whisker was selected for grey-value quantification. The image below is an enlargement of the red boxed section of the image above. (**C**) The grey values of the regrown whiskers treated with deoxyarbutin and control solvent were quantitatively measured, and the statistical differences were evaluated using a two-tailed paired Student’s *t*-test. *** *p* < 0.001. The scale bar of (**A**) is 2 mm and those of (**B**) are 1 and 0.2 mm.

**Figure 6 biomolecules-13-01076-f006:**
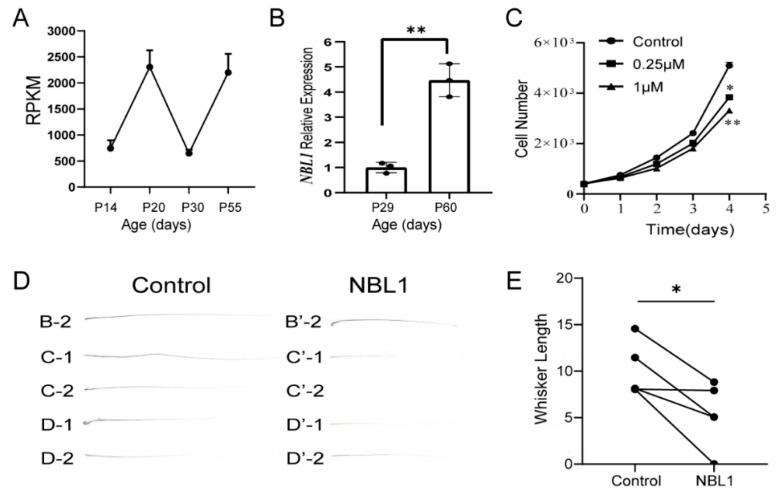
NBL1 is a negative hair-growth regulator. (**A**) Skin samples were harvested, and the expression profiles of the skin during the hair cycle were analyzed using RNA-seq (P14, developmental anagen; P20, 1st telogen; P30, 1st anagen; P55, 2nd telogen). The expression values of the *Nbl1* gene from the RNA-seq dataset were plotted. (**B**) Three pairs of samples from the back skin of mice at anagen (P29) and telogen (P60) were harvested, and the *Nbl1* expression was quantitatively analyzed using q-PCR. The statistical differences were evaluated using a two-tailed paired Student’s *t*-test. (**C**) Different recombinant NBL1 concentrations were added into culture media, and the growth curve of HaCaT cells was plotted over time. The statistical differences were evaluated using a two-tailed Student’s *t*-test. (**D**) Recombinant protein NBL1 and the control solvent were injected into five pairs of whisker follicles on the left and right sides on days 1, 3, and 5. On day 12, the regrown whiskers were plucked and imaged. (**E**) The lengths of regrown whiskers were quantitatively measured, and the statistical differences were evaluated using a Wilcoxon signed-rank test. *p* < 0.05 (*) and *p* < 0.01 (**).

## Data Availability

All data presented in this study are available in the main text, figures, and Appendix A.

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
