# Peer review of "Sensitive Quantitative In Vivo Assay for Evaluating the Effects of Biomolecules on Hair Growth and Coloring Using Direct Microinjections into Mouse Whisker Follicles"

_biomolecules, 2023, doi:10.3390/biom13071076_

Round 1

Reviewer 1 Report

Dear Authors

This manuscript is well-logically written. 

Here are comments

1. The number of references was too small, and the authors should increase to at least 40~50  references. In the introduction, many sentences had only one reference. In the discussion, there was no reference in spite of mentioning previous papers

2. Authors omitted several M&M including RNA-seq analysis, q-PCR, NBL1 experiment including HaCaT, and Coomassie brilliant blue experiments. The authors provided too simple M&M. The authors should provide all M&M in detail for understanding the relevant research scientists. 

3. Please, provide detailed information about steel microneedles, and NBL1 protein. 

4. Please, the authors emphasize the advantage of "Whisker Follicles Microinjection", and its scientific biological meaning or usage for future. 

Dear Authors

Please, provide the English editing service when the authors will submit a revised manuscript. 

Author Response

We very greatly appreciate the reviewers’ kind and professional comments and suggestions. Below we provide a point-by-point response of how the ms has been revised for every concern raised in our initial submission. All of the changes in the manuscript have been highlighted in yellow color. 

  1. The number of references was too small, and the authors should increase to at least 40~50  references. In the introduction, many sentences had only one reference. In the discussion, there was no reference in spite of mentioning previous papers

Reply: Thanks for the suggestions. We have added more relevant references to a total 47 in the manuscript.

  1. Authors omitted several M&M including RNA-seq analysis, q-PCR, NBL1 experiment including HaCaT, and Coomassie brilliant blue experiments. The authors provided too simple M&M. The authors should provide all M&M in detail for understanding the relevant research scientists. 

Reply: Thanks for the suggestions. We have added the methods of RNA-seq, q-PCR, growth curve, and the information of HaCaT and Coomassie brilliant in M&M section. For the details, please see the revised manuscript with tracked changes.

  1. Please, provide detailed information about steel microneedles, and NBL1 protein. 

Reply: Thanks for the suggestions. We have added the information on steel microneedles, and NBL1 protein in M&M section: micro-syringe with the needle (World Precision Instruments, Inc., NF36BL-2, outside diameter = 110 μm); NBL1 (50976-M08H) proteins were purchased from Sino Biological.

  1. Please, the authors emphasize the advantage of "Whisker Follicles Microinjection", and its scientific biological meaning or usage for future. 

Reply:In the “Discussion” section, we have addressed the advantage of the whisker follicle microinjection assay and its usage for future. To make it more clear, we modified the last paragraph as follows: “Overall, the whisker follicles microinjection assay we have established here is highly sensitive, quantitative, easy to operate and accurately reflects in vivo conditions. This nov-el assay can be used for investigating the functions of proteins or genes in relation to hair growth and coloring. Moreover, it serves as a valuable tool for evaluating the effects of biomolecules on hair growth and coloring. As a result, it will greatly aids in the develop-ment of these biomolecules into innovative drugs for treating diseases like hair loss and abnormal skin pigmentation.”

Please, provide the English editing service when the authors will submit a revised manuscript. 
Reply:The manuscript has been edited by the English editing service provided by MDPI.

Reviewer 2 Report

The current manuscript described an assay to evaluate protein functions and drug effects on hair growth and pigmentation by whisker follicle microinjection in mice. While the data look interesting and could be relevant for preclinical studies, the manuscript is expected to address the following concerns:

1). Although, as a specialized hair follicle, mouse whisker follicles share some characteristics with regular mouse hair follicles, there are certainly differences in biology and intracellular signaling between them. The author should provide the rationale and discuss why whisker follicles could be used for those assays to evaluate the functions or effects of proteins and agents on regular hairs.

2). There could be variability of whisker hair follicles in size. Thus, it is important to specify what size of the needles was used for microinjection.

3). For quantification and comparison, the two-tailed student’s t-test was applied. This is inappropriate for statistical analysis, such as in Figures 4B &D. I would recommend re-calculating the p values using an ANOVA test. Given the variability shown in Figure 6E, the Mann-Whitney U test is recommended.

4). Some minor issues of the English language, for example

In the title, mice whisker follicles could be mouse whisker follicles

Line 83-84, protein should be proteins.

 Some minor issues of the English language, for example

In the title, mice whisker follicles could be mouse whisker follicles

Line 83-84, protein should be proteins.

Author Response

We very greatly appreciate the reviewers’ kind and professional comments and suggestions. Below we provide a point-by-point response of how the ms has been revised for every concern raised in our initial submission. All of the changes in the manuscript have been highlighted in yellow color.

1). Although, as a specialized hair follicle, mouse whisker follicles share some characteristics with regular mouse hair follicles, there are certainly differences in biology and intracellular signaling between them. The author should provide the rationale and discuss why whisker follicles could be used for those assays to evaluate the functions or effects of proteins and agents on regular hairs.

Reply:Thanks for the suggestions. We have described the similarity and differences between whisker and regular hair and explained the rationale in the “Introduction” section:

“The mini-organ hair follicle is a skin appendage located deep in the dermis. A hair follicle consists of different compartments including the sebaceous gland, bulge, inner root sheath, outer root sheath, hair bulb, and dermal papilla. In hair bulbs, the hair matrix ep-ithelial cells proliferate to form the hair shaft, while whereas the melanocytes transport the melanosomes to the neighboring epithelial cells to deposit the pigments [33]. The dermal papilla regulates hair growth and coloring by releasing secretory factors [11, 33, 34].

The whisker, also called vibrissae, tactile hair, or sinus hair, is characterized by its follicle surrounded by two venous sinuses and a rigid fibrous capsule [35, 36]. Whisker follicles exhibit an anatomy and molecular signaling to that is similar to regular hair folli-cles, which explains why isolated whisker follicles isolated from mice have been em-ployed in culture to evaluate the effects of molecules on hair growth in preclinical research [14, 17, 18].”

2). There could be variability of whisker hair follicles in size. Thus, it is important to specify what size of the needles was used for microinjection.

Reply:Thanks for the suggestions. We have added the size of the needle and provided the catalog number and vender information of the needle (World Precision Instruments, Inc., NF36BL-2, outside diameter = 110 μm) in M&M section.

3). For quantification and comparison, the two-tailed student’s t-test was applied. This is inappropriate for statistical analysis, such as in Figures 4B &D. I would recommend re-calculating the p values using an ANOVA test. Given the variability shown in Figure 6E, the Mann-Whitney U test is recommended.

Reply:Thanks for the professional suggestions. For Figures 4B &D, we must compare the hair lengths between drug treatment and control groups in pairs. However, the ANOVA test could not meet the requirement of paired comparison. We actually set control for each dose of Minoxidil/Ruxolitinib treatment. To make this clearer, we have remade the graphs. We think that paired t-test is better than the ANOVA test in our case and that paired t-test can also reflect the information of dose effect. For Figure 6E, we used a Wilcoxon signed-rank test instead of Mann-Whitney U test to evaluate the significance of the difference, because we need to compare the data in pairs. We hope we didn’t misunderstand your meaning and purpose.

4). Some minor issues of the English language, for example

In the title, mice whisker follicles could be mouse whisker follicles

Line 83-84, protein should be proteins.

Reply:Thanks for pointing out the errors. We have sent our manuscript for professional English editing. The above and other mistakes have been corrected.

Round 2

Reviewer 1 Report

Dear Authors

The authors tried to revise their previous manuscript. 

But, it looked so complicated. 

Please revise your manuscript again by showing the changed contents in red and provide your responses to my feedback in another sheet. 

Don't use the memo in the manuscript. 

You should provide the complete form for a future publication.

Best Regards